Journal of Machine Learning Research ? (2025) 1-11    Submitted COMPAYL 2025; Published

# On the Importance of Text Preprocessing for Multimodal Representation Learning and Pathology Report Generation

**Ruben T. Lucassen ***                                      R.T.LUCASSEN@UMCUTRECHT.NL
*Dept. of Pathology, University Medical Center Utrecht, the Netherlands*
*Dept. of Biomedical Engineering, Eindhoven University of Technology, the Netherlands*

**Tijn van de Luijtgaarden ***
*Dept. of Mathematics and Computer Science, Eindhoven University of Technology, the Netherlands*

**Sander P. J. Moonemans**
*Dept. of Mathematics and Computer Science, Eindhoven University of Technology, the Netherlands*

**Gerben E. Breimer**
*Dept. of Pathology, University Medical Center Utrecht, the Netherlands*

**Willeke A. M. Blokx**
*Dept. of Pathology, University Medical Center Utrecht, the Netherlands*

**Mitko Veta**
*Dept. of Biomedical Engineering, Eindhoven University of Technology, the Netherlands*

## Abstract

Vision-language models in pathology enable multimodal case retrieval and automated report generation. Many of the models developed so far, however, have been trained on pathology reports that include information which cannot be inferred from paired whole slide images (e.g., patient history), potentially leading to hallucinated sentences in generated reports. To this end, we investigate how the selection of information from pathology reports for vision-language modeling affects the quality of the multimodal representations and generated reports. More concretely, we compare a model trained on full reports against a model trained on preprocessed reports that only include sentences describing the cell and tissue appearances based on the H&E-stained slides. For the experiments, we built upon the BLIP-2 framework and used a cutaneous melanocytic lesion dataset of 42,433 H&E-stained whole slide images and 19,636 corresponding pathology reports. Model performance was assessed using image-to-text and text-to-image retrieval, as well as qualitative evaluation of the generated reports by an expert pathologist. Our results demonstrate that text preprocessing prevents hallucination in report generation. Despite the improvement in the quality of the generated reports, training the vision-language model on full reports showed better cross-modal retrieval performance.

**Keywords:** Vision Language Modeling, Histopathology, Text Preprocessing

---

. * R.T. Lucassen and T. van de Luijtgaarden are co-first authors.

# 1 Introduction

Vision-language modeling has seen much improvement in recent years (Li et al., 2023; Liu et al., 2024; Radford et al., 2021; Yu et al., 2022). Following the success in the domain of natural images, similar models have been developed for medical domains such as pathology (Ahmed et al., 2024; Ding et al., 2024; Lu et al., 2024; Shaikovski et al., 2024). Applications of vision-language models in pathology include uni- and cross-modal retrieval of cases from databases and automated report generation. The latter can potentially alleviate the increasing workload of pathologists (Berbís et al., 2023; Van der Laak et al., 2021).

In addition to descriptions of cell and tissue appearances on hematoxylin and eosin (H&E)-stained whole slide images (WSIs), pathology reports often also include clinical information, patient history, and additional diagnostic results from immunohistochemical (IHC) stains and molecular tests. These types of information are either difficult to predict correctly or cannot accurately be inferred from the H&E-stained WSIs at all. If sentences with this information are part of the training dataset, then vision-language models are prone to hallucination (i.e., the generation of statements that contradict or cannot be verified from the source content) (Ji et al., 2023). For example, outcomes of IHC staining and molecular testing are present in the reports generated by PRISM (Shaikovski et al., 2024) and TITAN (Ding et al., 2024).

Although most of these errors can easily be recognized and corrected by a pathologist, doing so adds to the workload again, undermining the potential benefits of automation. To address this problem, Ahmed *et al.* (Ahmed et al., 2024) applied a post-processing procedure using regular expressions based on a set of keywords to remove specific information, such as the precise anatomical location, from generated reports. However, this approach increases the system's complexity for deployment, likely misses words that should be removed if the keyword set is not all-encompassing, and becomes more challenging to apply for removal of (sub)sentences.

Furthermore, unlike most pathology-specific vision-language models, which as of yet have been trained primarily to generate diagnoses (Ahmed et al., 2024; Shaikovski et al., 2024), our focus lies on generating descriptions of cell and tissue patterns. The motivation for this is twofold: (1) writing these descriptions is often the most time-consuming part for a pathologist and could, for that reason, yield the largest efficiency gain if automated; and (2) reaching a definitive diagnosis for ambiguous cases can be difficult, if not impossible, without the results from additional diagnostic tests.

As the main contribution, we investigate how selecting information from pathology reports for vision-language modeling affects the quality of the multimodal representations and generated reports. Building upon a text preprocessing workflow we developed in prior work (Lucassen et al., 2024b), we compare training on full pathology reports against training only on the H&E-related sentences that describe the cell and tissue appearances. All experiments were performed using the BLIP-2 framework (Li et al., 2023) and a dataset of H&E-stained WSIs with corresponding pathology reports for 19,636 cutaneous melanocytic lesions. Model performance was evaluated using image-to-text and text-to-image retrieval, as well as assessment of the accuracy and usability of the generated reports by an expert dermatopathologist. All code and model parameters are made publicly available [1].

---

1. https://github.com/nuldertien/PathBLIP-2

## 2 Materials and Methods

### 2.1 Dataset

The dataset used in this study consists of melanocytic lesion cases retrospectively collected from the digital archive of the Department of Pathology at the University Medical Center Utrecht, the Netherlands. All cases were accessioned between January 1, 2013, and December 31, 2020. More information about the curation process of the dataset can be found in (Lucassen et al., 2025). For each case, all unique, H&E-stained WSIs and the corresponding pathology report were included after de-identification. The study was conducted in compliance with the hospital's research ethics committee guidelines. Cases from patients who opted out of the use of their data for research purposes were excluded.

The pathology reports were preprocessed using a workflow described in detail in prior work (Lucassen et al., 2024b). After translation from Dutch to English, the reports were segmented into subsentences based on the information content. Pretrained language models were used for the translation and segmentation after being finetuned for the respective tasks. Two variants of each report were created by selecting part of the subsentences: (1) all sentences from the original report; and (2) only the sentences with cell and tissue appearances written based on the H&E-stained WSIs. All cases with an empty report for one or both of the variants were excluded from the dataset.

Acquisition of the WSIs was performed using either a ScanScope XT scanner (Aperio, Vista, CA, USA) at $20\times$ magnification with a resolution of 0.50 µm per pixel (slides scanned before 2016) or a NanoZoomer 2.0-XR scanner (Hamamatsu photonics, Hamamatsu, Shizuoka, Japan) at $40\times$ magnification with a resolution of 0.23 µm per pixel (slides scanned starting from 2016). To guide the WSI tessellation, tissue cross-sections and pen markings were segmented in each WSI at $1.25\times$ magnification using SlideSegmenter (Lucassen et al., 2024a). Non-overlapping tiles of $4{,}096\times4{,}096$ pixels were extracted from the WSIs at $20\times$ magnification. Tiles with identified pen markings or covered by tissue for less than 5% were excluded.

The dataset comprised of 42,433 H&E-stained WSIs from 19,636 melanocytic lesions with one report each, acquired from 14,951 unique patients. The majority of these lesions (81.9%) were benign common nevi, otherwise known as moles. The rest included non-common nevi, melanocytomas, and melanomas, ranging from benign to intermediate to malignant. In total, the reports describing only the H&E-related cell and tissue patterns contained 1,425,573 words across 129,121 sentences. In contrast, the full reports contained 2,132,008 words across 185,570 sentences. The dataset was split on a patient level into sets for training (80%), validation (10%), and testing (10%).

### 2.2 Vision-Language Model

We built upon the BLIP-2 framework (Li et al., 2023), which was designed for parameter-efficient vision-language modeling. An overview of the vision-language model and training procedure is shown in Fig. 1.

All extracted WSI tiles for a case are converted to 192-dimensional feature vectors using the second stage of HIPT (Chen et al., 2022) (i.e., two successive Vision Transformers (ViTs) (Dosovitskiy et al., 2021) pretrained on The Cancer Genome Atlas (TCGA)

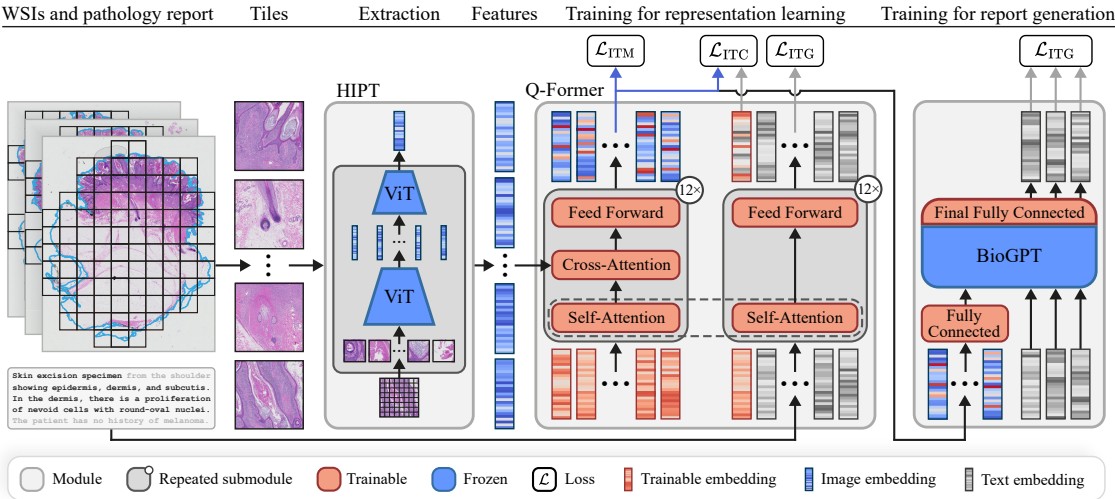

Figure 1: Overview of the vision-language modeling framework. Feature vectors are extracted using HIPT (Chen et al., 2022) from all tiles of the tessellated WSIs. Pathology reports, with or without information selection, are tokenized and embedded. The Q-Former is trained for representation learning in the first stage and report generation in the second stage.

dataset (Liu et al., 2018)). This image encoder is connected using the so-called Querying Transformer (Q-Former) to a pretrained language model with mostly frozen parameters. The Q-Former consists of two parallel Transformer (Vaswani et al., 2017) submodules of 12 blocks: (1) an image submodule with trainable embeddings of 768 dimensions as input, also referred to as query embeddings, that extract information from the image feature vectors using cross-attention layers in every other block; and (2) a submodule with tokenized text embeddings as input and without cross-attention layers. Both submodules share the same self-attention layers, which enables interaction between image and text information through the query embeddings.

The output query embeddings returned by the image submodule of the Q-Former are prepended to the sequence of tokens for autoregressive language generation using BioGPT (Luo et al., 2022). This language model has a decoder-only Transformer architecture with 24 layers, 347 million parameters, a vocabulary size of 42,384 tokens, and was pretrained on a biomedical text corpus.

## 2.3 Training for Representation Learning

To limit the required memory, only the Q-Former parameters were optimized for multimodal representation learning, while the parameters of the image encoder and text embedding layer remained frozen. The parameters of the Q-Former were initialized based on BERT-base-uncased (Devlin et al., 2019), except for the cross-attention layers and query embeddings, which were randomly initialized at the start of training. Following the first stage of the BLIP-2 training procedure (see Fig. 1), the Q-Former was trained using an

image-text contrastive loss $\mathcal{L}_{\text{ITC}}$, image-text matching loss $\mathcal{L}_{\text{ITM}}$, and image-grounded text generation loss $\mathcal{L}_{\text{ITG}}$.

The contrastive loss is optimized by maximizing the similarity between matching pairs of image and text embeddings, while minimizing the similarity between all unmatching pairs of image and text embeddings:

$$\mathcal{L}_{\text{ITC}} = -\frac{1}{2N} \left( \sum_{i}^{N} \log \frac{\exp(\max(\mathbf{x}_i^\top y_i)/\tau)}{\sum_{j=1}^{N} \exp(\max(\mathbf{x}_i^\top y_j)/\tau)} + \sum_{i}^{N} \log \frac{\exp(\max(y_i^\top \mathbf{x}_i)/\tau)}{\sum_{j=1}^{N} \exp(\max(y_i^\top \mathbf{x}_j)/\tau)} \right) \tag{1}$$

where $(\mathbf{x}_i, y_i)$ is the $i$-th matching image-text pair in the batch. Here, $\mathbf{x}_i$ are multiple, $L^2$-normalized output query embeddings from the Q-Former with image information, $y_i$ is a single text embedding for the [CLS] token, $N$ is the batch size, and $\tau$ is a trainable temperature parameter. A unimodal mask is used for the self-attention layers.

For the matching loss, the batch of matching image-text pairs is expanded with all images paired with an unmatching text, and all texts paired with an unmatching image, tripling the original batch size. Unmatching counterparts with a high similarity to the matching counterpart were more likely to be sampled. A fully connected layer is attached to predict, based on an output query embedding, whether an image-text pair matches or not. By averaging the predicted logits for all output query embeddings, a final prediction is obtained. The matching loss is optimized by minimizing the binary cross-entropy:

$$\mathcal{L}_{\text{ITM}} = -\frac{1}{3N} \sum_{i=1}^{3N} c_i \log P(\mathbf{q}_i) + (1 - c_i) \log(1 - P(\mathbf{q}_i)) \tag{2}$$

where $\mathbf{q}_i$ are the output query embeddings for the $i$-th image-text pair, $c_i$ is the binary label indicating whether the $i$-th pair matches or not, and $N$ is the original batch size. A bidirectional mask is used for the self-attention layers.

The image-grounded text generation loss is optimized by minimizing the cross-entropy of the paired text $\mathbf{y}$ under the forward autoregressive factorization with teacher-forcing (Williams and Zipser, 1989) to parallelize computation:

$$\mathcal{L}_{\text{ITG}} = -\frac{1}{T} \sum_{t=1}^{T} \log P(y_t|\mathbf{y}_{1:t-1}, \mathbf{x}) \tag{3}$$

where $y_t$ is the $t$-th token of the text report $\mathbf{y}$ with length $T$, $\mathbf{y}_{1:t-1}$ represents all embedded tokens preceding the embedding of the current token, and $\mathbf{x}$ represents the query embeddings with image information from the Q-Former. A multimodal, causal mask is used for the self-attention layers.

The model with 16 querying embeddings was trained on the sum of $\mathcal{L}_{\text{ITC}}$, $\mathcal{L}_{\text{ITM}}$, and $\mathcal{L}_{\text{ITG}}$ for 25 epochs with a batch size of 20. Model parameters were updated using a learning rate of $1 \cdot 10^{-4}$ with a cosine learning rate scheduler and 1,000 warmup steps. The AdamW (Loshchilov and Hutter, 2019) optimization algorithm ($\beta_1 = 0.9$, $\beta_2 = 0.999$) was used with weight decay equal to 0.01. The model was trained with label smoothing (Szegedy et al., 2016) for the contrastive loss with $\alpha = 0.9$ (which was omitted from Eq. 1 for brevity). Hyperparameters were tuned based on the validation set results. The final model parameters were selected based on the epoch with the lowest validation loss.

### 2.4 Training for Report Generation

In the second stage of the BLIP-2 training procedure, the Q-Former is connected to BioGPT for report generation (see Fig. 1). A fully connected layer is used to transform the output embeddings with image information from the Q-Former to the dimensionality of the text embeddings. The parameters of the Q-Former (pretrained in the prior representation learning stage using full reports), the fully connected layer, and the final fully connected layer of the BioGPT model were optimized during training based on the image-grounded text generation loss in Eq. 3.

The model with 64 querying embeddings was trained for 21 epochs with a batch size of 36. Model parameters were updated using a learning rate of $1 \cdot 10^{-3}$ with a cosine learning rate scheduler and 1,000 warmup steps. The AdamW (Loshchilov and Hutter, 2019) optimization algorithm ($\beta_1 = 0.9$, $\beta_2 = 0.999$) was used with weight decay equal to 0.01. Hyperparameters were tuned based on the validation set results. The final model parameters were selected based on the epoch with the lowest validation loss.

## 3 Results

### 3.1 Retrieval Performance

The quality of the learned representations was evaluated using cross-modal retrieval based on the cases in the independent test set ($N = 1{,}970$). Cross-modal retrieval assesses to what extent the pathology reports can be matched to the corresponding WSIs (and vice versa) based on the similarity of the image and text representations. Performance on the retrieval tasks was expressed in terms of the recall at $k$ (i.e., the proportion of cases for which the matching item is in the top $k$ retrieved items), as well as the mean and median rank of the matching items retrieved. Bootstrapping ($R = 1{,}000$ samples) was used to calculate 95% confidence intervals (CIs) using the percentile method. The set of items for retrieval was not sampled as part of the bootstrapping procedure to prevent matching conflicts for duplicates. The retrieval performance was evaluated for the models trained on the two report variants (i.e., full reports or only the descriptions of H&E-related cell and tissue patterns) and repeated using both variants of the reports for retrieval.

The results for the cross-modal retrieval are shown in Table 1 for image-to-text matching and in Table 2 for text-to-image matching. The best retrieval performance was achieved by the model trained on full reports when the full reports were also used for matching. The two models performed on par when matching was done using the preprocessed reports including only the sentences with cell and tissue appearances based on H&E-stained WSIs. The worst retrieval scores were seen for the model trained on the preprocessed reports when the full reports were used for matching. Text-to-image matching performed notably worse than image-to-text matching for this combination of training and retrieval. This is in contrast to all other combinations, where the results for image-to-text and text-to-image matching were comparable.

### 3.2 Report Generation Performance

We performed a reader study to evaluate the quality of the generated reports. A total of 50 cases from the test set were randomly selected with stratification based on the diagnosis (25

Table 1: Results for image-to-text matching based on the cases in the independent test set ($N = 1{,}970$). Note that lower scores represent better performance for the rank.

| Training | Retrieval | Recall at $k$ | | | Rank | |
| --- | --- | --- | --- | --- | --- | --- |
| | | $k=1$ | $k=5$ | $k=10$ | Mean | Median |
| H&E only | H&E only | 0.022 | 0.105 | 0.176 | 137.7 | 61 |
| | | (0.017-0.030) | (0.092-0.119) | (0.159-0.192) | (128.8-146.7) | (56-67) |
| | Full report | 0.018 | 0.081 | 0.131 | 172.4 | 79 |
| | | (0.013-0.024) | (0.068-0.091) | (0.118-0.145) | (162.8-182.6) | (72-88) |
| Full report | H&E only | 0.024 | 0.095 | 0.164 | 135.7 | 63 |
| | | (0.017-0.031) | (0.082-0.107) | (0.147-0.180) | (127.5-144.5) | (56-68) |
| | Full report | 0.058 | 0.182 | 0.279 | 83.3 | 31 |
| | | (0.048-0.068) | (0.164-0.198) | (0.259-0.300) | (77.4-89.5) | (29-35) |

Table 2: Results for text-to-image matching based on the cases in the independent test set ($N = 1{,}970$). Note that lower scores represent better performance for the rank.

| Training | Retrieval | Recall at $k$ | | | Rank | |
| --- | --- | --- | --- | --- | --- | --- |
| | | $k=1$ | $k=5$ | $k=10$ | Mean | Median |
| H&E only | H&E only | 0.027 | 0.104 | 0.168 | 141.0 | 60.5 |
| | | (0.019-0.034) | (0.090-0.116) | (0.150-0.184) | (131.8-150.3) | (56-67) |
| | Full report | 0.014 | 0.064 | 0.113 | 202.3 | 101.5 |
| | | (0.009-0.019) | (0.053-0.074) | (0.099-0.127) | (190.9-214.2) | (92-109) |
| Full report | H&E only | 0.024 | 0.090 | 0.154 | 138.6 | 62 |
| | | (0.017-0.030) | (0.077-0.103) | (0.138-0.170) | (130.4-146.7) | (57-67) |
| | Full report | 0.058 | 0.172 | 0.270 | 87.0 | 31 |
| | | (0.047-0.068) | (0.155-0.188) | (0.250-0.288) | (80.4-93.0) | (29-35) |

common nevi and 25 melanocytic lesions of various other subtypes) to cover both common and rare entities. For all selected cases, the original report written by a pathologist as part of routine clinical practice was collected and the two vision-language model variants were used to generate a report. A pathologist (W.B.) experienced in dermatopathology was recruited to independently evaluate the three reports per case. The evaluation consisted of counting factual errors, unverifiable statements, important missing information, and repeated phrases. The reports were also scored on a scale from 1 to 5, ranging from mostly inaccurate with no expected benefit from using the report as starting point to highly accurate with minimal to no adjustments needed for use in clinical practice. The pathologist only had access to the WSIs during the reader study. To prevent bias in the evaluation, reports were randomly ordered per case and the pathologist was blinded from the origin of the report.

The mean and standard deviation of the error counts and quality score from the reader study are shown in Table 3. The vision-language model trained on full reports produced, on average, $3.0\,(\pm 4.8)$ statements that could not be verified based on the H&E-stained WSIs per report. Less unverifiable statements were produced for common nevi, averaging $1.2\,(\pm 0.4)$ occurrences compared to $4.9\,(\pm 6.4)$ for other subtypes of melanocytic lesions. This is in line with the proportion of the reports that does not describe H&E-related cell

Table 3: Results of the reader study with blinded evaluation by a pathologist are presented as the mean and standard deviation. The score reflects the overall accuracy and usability of the reports on a 1-5 scale. The count of unverifiable statements is not applicable to pathologist-written reports.

| Data | Written by | Error count per report | | | | Score |
|---|---|---|---|---|---|---|
| | | Factual | Unverifiable | Omission | Repetition | |
| All | Model - Full report | 1.9 ± 1.8 | 3.0 ± 4.8 | 0.2 ± 0.5 | 0.8 ± 2.1 | 2.4 ± 1.2 |
| (N = 50) | Model - H&E only | 1.7 ± 1.7 | 0.0 ± 0.0 | 0.6 ± 1.1 | 0.1 ± 0.5 | 2.8 ± 1.4 |
| | Pathologist | 0.7 ± 1.0 | − | 0.3 ± 0.5 | 0.8 ± 1.8 | 3.9 ± 1.0 |
| Common nevi | Model - Full report | 1.0 ± 1.0 | 1.2 ± 0.4 | 0.2 ± 0.5 | 0.0 ± 0.0 | 3.2 ± 1.2 |
| (N = 25) | Model - H&E only | 0.9 ± 0.9 | 0.0 ± 0.0 | 0.3 ± 0.5 | 0.1 ± 0.3 | 3.7 ± 1.2 |
| | Pathologist | 0.5 ± 0.7 | − | 0.3 ± 0.5 | 0.1 ± 0.3 | 4.2 ± 0.9 |
| Other lesions | Model - Full report | 2.8 ± 2.0 | 4.9 ± 6.4 | 0.2 ± 0.4 | 1.5 ± 2.8 | 1.7 ± 0.7 |
| (N = 25) | Model - H&E only | 2.5 ± 1.9 | 0.0 ± 0.0 | 1.0 ± 1.5 | 0.1 ± 0.6 | 1.9 ± 1.0 |
| | Pathologist | 0.9 ± 1.3 | − | 0.3 ± 0.5 | 1.5 ± 2.3 | 3.6 ± 1.0 |

and tissue patterns in the dataset, with and average of 25.3% of the words for common nevi and 43.0% of the words for other melanocytic lesion subtypes. Additionally, more repeated information was seen in the reports generated by the model trained on full reports. In comparison, no statements were generated that could not be supported nor contradicted based on the H&E-stained WSIs by the model with preprocessed reports as training data. The number of factual errors was comparable for the two vision-language models. Overall, the accuracy and usability of the reports generated by both models was scored higher for common nevi than for the other melanocytic lesions. The scores for the reports written by pathologists as part of routine clinical practice were the highest, although considerable inter-observer disagreement was seen as well, based on the average of $0.7\,(\pm 1.0)$ factual errors in these reports.

## 4 Discussion and Conclusion

Our study investigated the effect of information selection as part of text preprocessing on the performance of vision-language models in pathology. In cross-modal retrieval, the model trained on full pathology reports outperformed the model trained only on the descriptions of H&E-related cell and tissue patterns. Studying which types of additional information contribute positively to retrieval performance is an interesting direction for future work. Models trained on full reports, however, were also prone to generating unverifiable as well as redundant statements, particularly for more uncommon melanocytic lesions. Despite the unique characteristics of each pathology domain, we expect that these results generalize beyond melanocytic skin lesions.

In conclusion, our findings suggest that text preprocessing effectively prevents hallucination in pathology report generation. While this improved the overall accuracy and usability of generated reports, albeit not yet to the level of a pathologist, training on full reports showed superior performance in cross-modal retrieval.

## Acknowledgments and Disclosure of Funding

This research was financially supported by the Hanarth Foundation.

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
