# OpenReview forum: "On the Importance of Text Preprocessing for Multimodal Representation Learning and Pathology Report Generation"
_MICCAI.org/2025/Workshop/COMPAYL — COMPAYL 2025_

### Official Review · Reviewer_xmvD · 2025-07-13
**On the Importance of Text Preprocessing for Multimodal Representation Learning and Pathology Report Generation**

**Rating:** 2
**Confidence:** 4

**Review:**

## Summary
The authors investigate the impact of using pre-processed pathology reports for vision-language model pre-training on the performance of cross-modal retrieval and report generation tasks. The study compares models trained on full reports versus those trained on pre-processed reports to assess improvements in accuracy, hallucination reduction, and overall utility.

- **Clarity**: The paper is generally clear but lacks sufficient detail on preprocessing steps.
- **Quality**: The study is well-structured with reproducible results but suffers from weak evaluation.
- **Originality**: The model design is novel.
- **Significance**: The findings are incremental, with limited real-world clinical impact.

## Strengths
- **Novel and Interesting Model Design**: The proposed vision-language model architecture is innovative, offering a novel approach to integrating textual and visual data in pathology.
- **Detailed Hyperparameter Description**: The authors provide a thorough explanation of the hyperparameters used during model training, which enhances reproducibility.
- **Open-Source Code and Parameters**: The availability of the code and model parameters facilitates independent validation and further research.
- **Reduction of Hallucinations**: The preprocessing method effectively diminishes hallucinations in generated reports, a critical improvement for clinical applications.

## Weaknesses
- **Insufficient Preprocessing Description**: The paper lacks a detailed explanation of the preprocessing steps, particularly regarding the effort required. Quantifying the time, computational cost, or manual labor involved would strengthen the study by demonstrating the trade-off between preprocessing effort and performance gains.
- **Weak Evaluation Methodology**: The evaluation could be improved by incorporating external validation, such as testing on publicly available datasets like TCGA, where both H&E slides and corresponding reports are accessible. This would better assess the method’s generalizability and robustness.
- **Underwhelming Results**: The findings are somewhat disappointing. While preprocessing improves report generation compared to the full-report baseline, the performance remains far below expert-level accuracy, raising questions about the practical utility of the method. Additionally, preprocessing does not enhance cross-modal retrieval, which is a surprising and concerning result.

---

### Official Review · Reviewer_1ArY · 2025-07-15
**smart approach to pathology report generation**

**Rating:** 4
**Confidence:** 4

**Review:**

The article presents a really interesting idea of removing part of the content of pathology reports that are not related to the actual image content but rather the context in which the images were taken. This can reduce the errors linked toassociatin some of the context with visual features in the images.
The paper is well written and understandable. The evaluation is not trivial but is done here with a real user test, which is very good! There is definitely some subjectivity and the set for the evaluation is small but the results seem to make clear sense.
It would be interesting to have more details on how the reports are translated and whether the translation has an effect on the quality of the reports generated or the evaluation. It woudl also be good add more details on how sentences were selected. Was this partly manual? Or trained? How completely is this part and what is the influence of this on the final results.
Otherwise the described techniques make sense, and I believe that paper would be interesting for the workshop.

---

### Official Review · Reviewer_eujn · 2025-07-21
**Impact of Text Preprocessing in VLMs**

**Rating:** 4
**Confidence:** 4

**Review:**

This paper presents a well-executed study on the impact of text preprocessing in vision-language modelling for pathology, specifically in the context of report generation and cross-modal retrieval. The authors compare models trained on full pathology reports versus those trained on pre-processed reports containing only H&E-related descriptions. The work is timely, relevant, and contributes meaningfully to the field of computational pathology.

Pros
1. The focus on text preprocessing to reduce hallucinations in generated pathology reports is novel and addresses a critical issue in clinical deployment.
2. The methodology is rigorous, leveraging the BLIP-2 framework and a large, well-curated dataset of 42,433 WSIs and 19,636 reports.
3. The paper is well-written and clearly structured, with thorough explanations of the preprocessing pipeline and evaluation metrics.
4. The findings have practical implications for improving the reliability of automated report generation in pathology.
5. Code and model parameters are made publicly available, supporting reproducibility.

Cons
1. While preprocessing improves report quality, it slightly compromises retrieval performance. A deeper analysis of this trade-off would strengthen the discussion.
2. The study focuses on melanocytic lesions; broader validation across other pathology domains would enhance impact.
3. The reader study is limited to 50 cases. Expanding this could provide more robust insights into clinical applicability.